# Effect of lifting COVID-19 restrictions on utilisation of primary care services in Nepal: a difference-in-differences analysis

Neena R Kapoor 🄳 ,[1] Amit Aryal,[2] Suresh Mehata,[3] Mahesh Dulal,[4] Margaret E Kruk,[1] Sebastian Bauhoff,[1] Catherine Arsenault[1]

¹Department of Global Health and Population, Harvard T. H. Chan School of Public Health, Boston, Massachusetts, USA
²Swiss TPH, University of Basel, Basel, Switzerland
³Ministry of Health, Government of Nepal, Biratnagar, Province 1, Nepal
⁴Office of the Member of Federal Parliament Nepal, Gagan Kumar Thapa, Kathmandu, Nepal

**Correspondence to**
Neena R Kapoor;
nkapoor@hsph.harvard.edu

## ABSTRACT

**Introduction** An increasing number of studies have reported disruptions in health service utilisation due to the COVID-19 pandemic and its associated restrictions. However, little is known about the effect of lifting COVID-19 restrictions on health service utilisation. The objective of this study was to estimate the effect of lifting COVID-19 restrictions on primary care service utilisation in Nepal.

**Methods** Data on utilisation of 10 primary care services were extracted from the Health Management Information System across all health facilities in Nepal. We used a difference-in-differences design and linear fixed effects regressions to estimate the effect of lifting COVID-19 restrictions. The treatment group included palikas that had lifted restrictions in place from 17 August 2020 to 16 September 2020 (Bhadra 2077) and the control group included palikas that had maintained restrictions during that period. The pre-period included the 4 months of national lockdown from 24 March 2020 to 22 July 2020 (Chaitra 2076 to Ashar 2077). Models included month and palika fixed effects and controlled for COVID-19 incidence.

**Results** We found that lifting COVID-19 restrictions was associated with an average increase per palika of 57.5 contraceptive users (95% CI 14.6 to 100.5), 15.6 antenatal care visits (95% CI 5.3 to 25.9) and 1.6 child pneumonia visits (95% CI 0.2 to 2.9). This corresponded to a 9.4% increase in contraceptive users, 34.2% increase in antenatal care visits and 15.6% increase in child pneumonia visits. Utilisation of most other primary care services also increased after lifting restrictions, but coefficients were not statistically significant.

**Conclusions** Despite the ongoing pandemic, lifting restrictions can lead to an increase in some primary care services. Our results point to a causal link between restrictions and health service utilisation and call for policy makers in low- and middle-income countries to carefully consider the trade-offs of strict lockdowns during future COVID-19 waves or future pandemics.

## BACKGROUND

In a time of crisis, high-quality health systems have two tasks: respond to the crisis and maintain the provision and quality of essential health services.[1] Health systems in low-income

---

### STRENGTHS AND LIMITATIONS OF THIS STUDY

⇒ We included data on 10 wide-ranging primary care services extracted from the Nepal Health Management Information System (HMIS).

⇒ We used a difference-in-differences design to compare service use in palikas that lifted restrictions to those that maintained them, which controls for time-fixed differences between palikas and temporal trends common to both groups.

⇒ We controlled for new COVID-19 cases at the district level, but other time-varying confounders could affect the two groups differently.

⇒ HMIS data provide real-time information on patterns in service use however, despite the data cleaning conducted, data quality issues and underreporting by some facilities could bias our results.

---

countries, which may already be underfunded, under-resourced and overburdened, may be particularly vulnerable during the COVID-19 pandemic. An increasing number of studies have reported disruptions in health service utilisation since the start of the pandemic in low- and middle-income countries (LMICs).[2–10] The ongoing COVID-19 pandemic has directly strained healthcare systems around the world that are struggling to meet the physical resource, human resource (numbers and skills) and service coordination demands of the pandemic. The pandemic may also have had indirect effects on primary healthcare utilisation, as restrictions and lockdowns implemented by governments to reduce the spread of COVID-19 may affect people's ability or willingness to visit healthcare facilities.

Nepal is a lower-middle-income country of South Asia with a population of 28.6 million.[11] The country has shown significant gains in health and healthcare utilisation over the past decade. The pandemic could reverse these hard-won gains. As of December

2021, COVID-19 had infected more than 800 000 people across Nepal and had led to a reported 11 594 deaths.[12] Despite the existence of effective vaccines, only 32.8% of the Nepali population is currently fully vaccinated against COVID-19.[12]

Following the declaration of the pandemic on 11 March 2020 by the WHO, healthcare utilisation declined substantially in Nepal, ranging from a 65% decline in tuberculosis (TB) case detection to a 4% decline in contraceptive use.[3] Many factors may be responsible for a decline in health service utilisation during the pandemic. Declines may stem from the pandemic itself (perceived threat), the actual number of new COVID-19 cases reported in a given period (leading to a fear of infection when visiting facilities or to overburdened health facilities treating COVID-19 patients) or from the restrictions imposed (ie, lockdowns) to curb the spread of COVID-19. The barriers imposed by COVID-19 restrictions, such as stay-at home requirements or public transport closures, may play an important role in affecting healthcare utilisation. In Madesh Pradesh (formerly known as province 2 of Nepal), people reported that the national lockdown restricted accessibility to health facilities and deterred them from seeking care.[13] An increasing number of studies have described the effects of the pandemic and associated restrictions on healthcare utilisation.[2–10] However, little is known about the effect of lifting COVID-19 restrictions on healthcare utilisation. Understanding these effects is crucial to plan for potential rebounds in demand and determining whether potentially weakened health systems can cope with surges in demand.

In the wake of the COVID-19 pandemic, in March 2020, the Government of Nepal implemented a countrywide lockdown.[14 15] After almost 4 months of strict lockdown, in July 2020, the decision was made to end the national lockdown and lift most of these restrictions at the national level.[16] However, some of Nepal's municipality governments decided to maintain restrictions to contain the spread of COVID-19. This contrast in removal of restrictions gave rise to a natural experiment that allowed us to estimate the causal effect of lifting COVID-19 restrictions on healthcare utilisation.

In this study, we used a difference-in-differences design (DID) to estimate the effect of lifting COVID-19 restrictions on primary care service utilisation in Nepal. Understanding the effect of lifting COVID-19 restrictions on primary healthcare is crucial to inform policy responses during future waves of COVID-19 or future pandemics in LMICs.

## METHODS
### Data sources
We used data from the Nepal Health Management Information System (HMIS) obtained through the DHIS2 platform, formerly known as the district health information system-2. The HMIS in Nepal includes information from all health facilities in the country including both public and private facilities across all levels of the health system.[17] A total of 7605 health facilities are expected to report to the DHIS2 across 753 urban and rural municipalities known as 'palikas' (palikas are a local form of government in Nepal's federal system).

Information on the types of COVID-19 restrictions in place was obtained from various sources including: INSECOnline, a human rights news portal in Nepal providing daily COVID-19 updates, the Nepal COVID-19 Crisis Management Coordination Center (CCMCC) government sites, District Administration Office (DAO) sites and additional online news sources (online supplemental table 1).[18–20]

We also included data on the total number of COVID-19 cases at district level in Nepal (COVID-19 case counts were not available at the palika level). Monthly COVID-19 cases in each of the 77 districts were obtained from the Nepal Health Emergency Operation Centre, Ministry of Health and Population.[21]

### Measures

#### Primary care service utilisation
We aimed to include 12 primary care services: contraceptive users, antenatal care (ANC) visits, postnatal care (PNC) visits, visits for children under 5 with pneumonia, visits for children under 5 with diarrhoea, pentavalent vaccinations, measles vaccinations, visits for diabetes, visits for hypertension, number of HIV tests conducted, number of TB cases detected and total outpatient visits. Selection of these services was based on availability in the DHIS2 and because they covered a range of health needs including sexual, reproductive, maternal, newborn, child, and adolescent health services, infectious diseases, and non-communicable diseases. Detailed definitions are in online supplemental table 2.

We obtained the monthly number of each of these services provided from 15 January 2019 to 13 January 2021 (Nepali calendar Magh 2075 to Poush 2077). These data were available at the palika level.

Because DHIS2 data are self-reported by health facilities, these data may contain errors. Our data cleaning procedures entailed identifying positive outliers (greater than 3.5 SD from the mean trend) and setting any outliers as missing.[22] We did not assess negative outliers since decreases in utilisation were expected during the lockdown period. For each health service, we also excluded palikas that were missing any data during the 5-month study period (a complete-case analysis).

#### COVID-19 restrictions
From 24 March 2020 to 22 July 2020, the Federal Government of Nepal imposed a strict nationwide lockdown in response to the pandemic. This included stay-at-home requirements except for essential services, businesses, public transport and school closures, and restrictions on large gatherings, international travel and internal movement (see online supplemental table 3). On July 22, major

restrictions were lifted at the national level, including stay-at-home requirements, non-essential business and public transport closures, but some districts and palikas maintained these restrictions. Following the lifting of the national lockdown, 248 palikas lifted the restrictions while 505 palikas maintained one or more of these restrictions.

For this analysis, the treatment group includes the palikas that lifted these restrictions, while the control group includes palikas that continued at least one or more restriction. The pre-intervention period includes the 4 months from 14 March to 15 July (which corresponds to the Nepali months of Chaitra 2076 to Ashar 2077) and the post-intervention period is 17 August 2020 to 16 September 2020 (the Nepali month of Bhadra 2077). 16 July 2020 to 16 August 2020 (Shrawan 2077) was excluded from the analysis since lifting of the national lockdown occurred mid-month. Our analysis used Nepal calendar months as the unit of time.

The classification of palikas into treated and control groups was done using primarily INSEConline, an online news portal that provided daily updates on the COVID-19 situation in Nepal.[18] Four of the coauthors extracted information on the types of restrictions in place in each palika from the INSEConline news reports and verified and complemented the information with CCMCC government sites, DAO sites and additional news sources (online supplemental table 1).[19 20] Any disagreements were resolved through discussion. We used a 10-day threshold as a general rule of thumb. If restrictions were in place for less than 10 days during the month, the palika was classified as having lifted the restrictions and was included in the treatment group. If the restrictions covered more than 10 days, the palika remained in the control group (maintained restrictions). However, given imprecision in some of the policy reports, it was not always possible to apply this threshold with precision in some palikas.

Figure 1 shows the timeline of COVID-19 restrictions and cases in Nepal from 1 January 2020 to 16 September 2020. The first COVID-19 case was reported in Nepal on 25 January 2020. Notably, the end of the national lockdown on 22 July 2020 coincided with the beginning of the first real COVID-19 wave (figure 1).

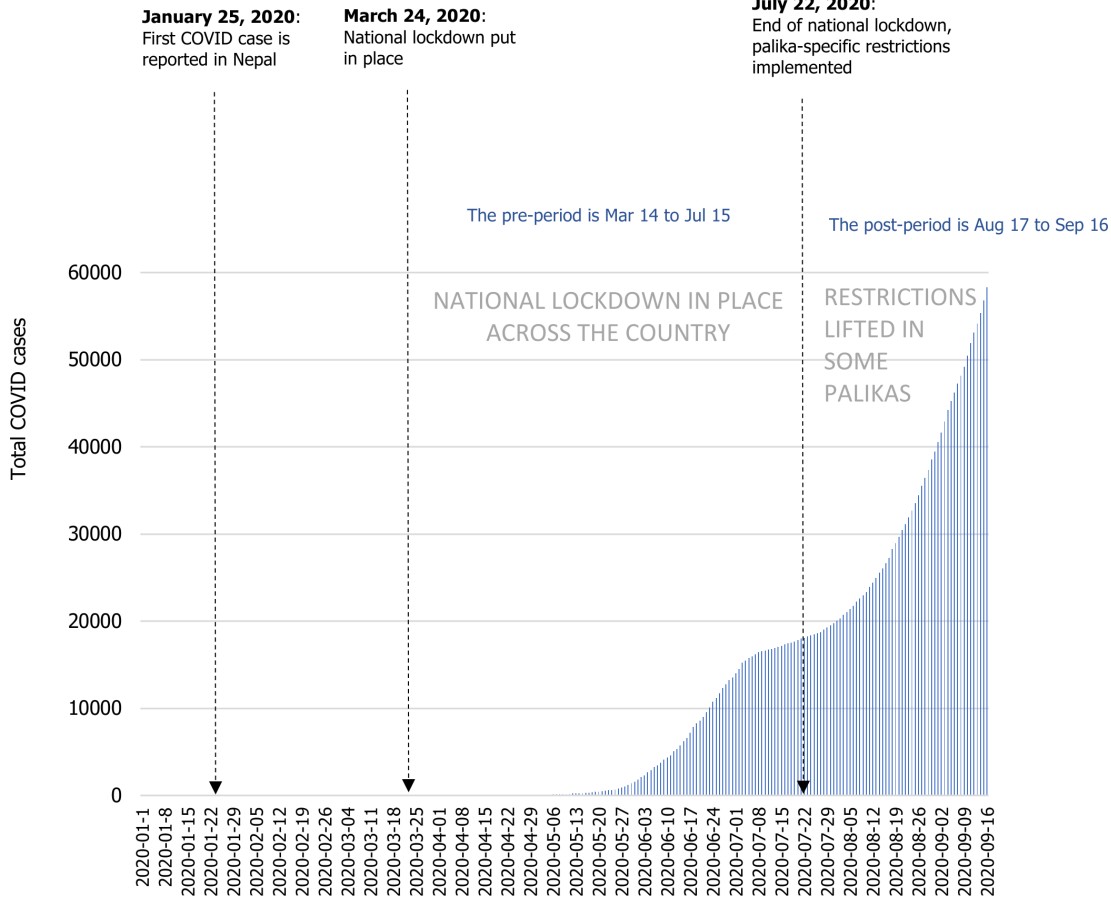

**Figure 1** Total COVID-19 cases and policy responses in Nepal from 1 January 2020 to 16 September 2020. The first recorded COVID-19 case was reported on 25 January 2020. The federal government implemented a nationwide lockdown on 24 March 2020, including: stay-at-home requirements, closure of non-essential businesses, schools and all public transport, and restrictions on gathering and internal movements.[14] The National lockdown was lifted 4 months later on 22 July 2020, with major restrictions lifted, including stay-at-home requirements, workplace and public transport closures after which palika-specific response was allowed. Source: COVID-19 Data Repository by the Centre for Systems Science and Engineering at Johns Hopkins University.[38]

## Statistical analysis

The analysis was conducted at the palika level, using Nepali calendar months as the unit of time. DID analysis is often used in policy evaluations to compare outcomes before and after a policy change for a group affected by the change (treated group) to a group not affected by the change (control group).[23] We used a DID design and fixed effects ordinary least square regression models. The following model was used and repeated for each of the health services analysed:

$$S_{pt} = \alpha + \beta \left[ lifted\ lockdown_{pt} \right] + \gamma_t + \delta_p + X_{dt} + \varepsilon_{pt}$$

Where $S_{pt}$ is the number of health services (number of visits or users) provided in palika $p$ in month $t$, $\gamma_t$ and $\delta_p$ are vectors of month and palika fixed effects, respectively, and $X_{dt}$ is the number of new COVID-19 cases in district $d$ and month $t$. The coefficient of interest is $\beta$, which represents the difference in service utilisation among palikas that lifted restrictions compared with those that maintained restrictions. The palika fixed effects controls for time-invariant differences between palikas and avoids the need to control for time-fixed confounders. For example, the palika fixed effects will control for unmeasured differences between palikas (urbanicity, population size, wealth) that can affect service utilisation. The DID design also controls for all factors commonly affecting the outcomes in all palikas over time, through month fixed effects. COVID-19 incidence would be associated with both the exposure (restrictions), and the outcome (health service utilisation) and may vary between the treatment and control groups. Thus, we included monthly COVID-19 cases in the regression models to control for potential confounding. Models also included clustered standard errors at the palika level.

A main assumption of DID models is that the outcome trend in the control group represents a good approximation of what the outcome trend would have been in the treatment group in the absence of the policy change (ie, the counterfactual trend). Thus, to probe the assumption that the control palika trends were a good counterfactual for the treatment group (palikas that lifted restrictions), we implemented a series of tests. First, we conducted a pretrend placebo test by comparing the difference in service utilisation between the treated and control palikas in May and June 2020 (Jestha and Ashar 2077) compared with April 2020 (Baisakh 2077). We performed a joint F test of whether these coefficients were jointly zero (online supplemental materials). Since all palikas were under the same restrictions from March to June 2020 (Chaitra 2076 to Ashar 2077), there should be no effect, providing evidence for parallel trends in the pre-period. Second, we assessed the parallel trend assumption graphically (figure 2). We also conducted a sensitivity analysis, excluding 14 March 2020 to 12 April 2020 (Chaitra 2076) from the analysis, since the national lockdown was put in place in the middle of this month, to see if the results differed.

### Patient and public involvement

Patients will be involved in dissemination of this research. There was no patient or public involvement in the design, reporting or interpretation of results.

## RESULTS

The average number of services provided during the national lockdown (pre-intervention period) and after the national lockdown was lifted (post-intervention period) for treated and control palikas are shown in table 1. Table 1 also shows the number of palikas included in the analysis of each health service. Service utilisation tended to be lower in the treatment group for most services. The palikas in the treatment group, those that lifted restrictions, had fewer COVID-19 cases and smaller

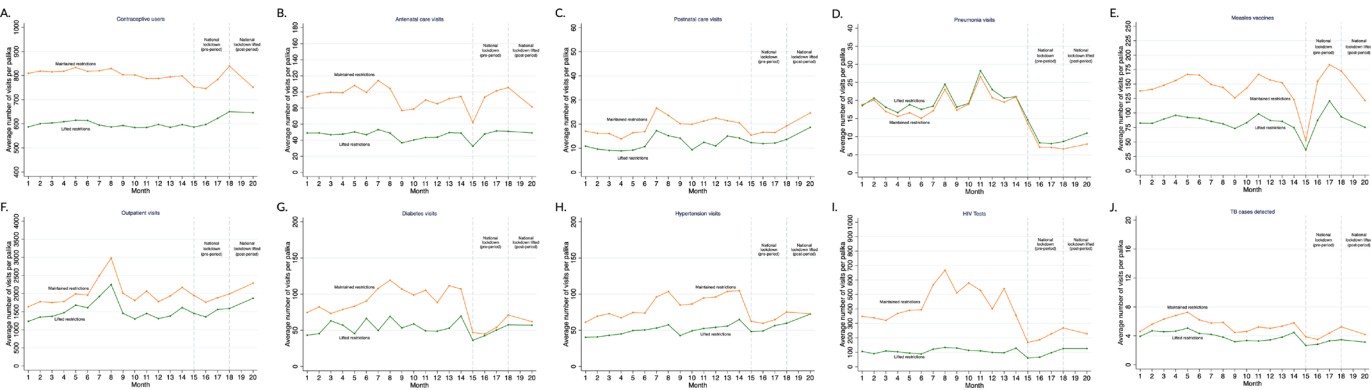

**Figure 2** Primary care service utilisation from 15 January 2019 to 16 September 2020 (Magh 2075 to Bhadra 2077 in Nepal). Months 1–20 are 15 January 2019 to 16 September 2020 (Magh 2075 to Bhadra 2077). For the purpose of our analysis, the national lockdown period (pre-period) includes months 15–18 (14 March 2020 to 15 July 2020, Chaitra 2076 to Ashar 2077) and the post-period is month 20 (17 August 2020 to 16 September 2020, equivalent to Bhadra 2077). The orange lines represent average healthcare utilisation in the control group: palikas that maintained COVID-19 restrictions in the post-period (eg, stay-at-home requirements, business and public transport closures). The green lines represent healthcare utilisation in the treatment group: palikas that lifted COVID-19 restrictions in the post-period. Detailed definitions of health service indicators are in online supplemental materials.

**Table 1** Number and characteristics of palikas by treatment group and period

| | Treatment group (lifted restrictions) | | Control group (maintained restrictions) | |
| --- | --- | --- | --- | --- |
| | Pre-period | Post-period | Pre-period | Post-period |
| | Average per month (N) | Average per month (N) | Average per month (N) | Average per month (N) |
| Contraceptive users | 614.09 (245) | 646.10 (245) | 780.52 (497) | 751.41 (497) |
| ANC visits | 45.66 (235) | 48.97 (235) | 90.57 (490) | 81.52 (490) |
| PNC visits | 12.49 (182) | 18.65 (182) | 16.93 (337) | 24.63 (337) |
| Child pneumonia visits | 9.92 (186) | 10.98 (186) | 8.59 (369) | 7.95 (369) |
| Measles vaccine | 84.47 (102) | 75.28 (102) | 140.83 (246) | 121.74 (246) |
| Outpatient visits | 1489.23 (243) | 1870.75 (243) | 1896.25 (499) | 2286.77 (499) |
| Diabetes visits | 46.66 (100) | 57.04 (100) | 54.27 (233) | 62.00 (233) |
| Hypertension visits | 53.43 (220) | 72.44 (220) | 65.50 (469) | 72.93 (469) |
| HIV tests | 87.55 (68) | 125.88 (68) | 212.92 (168) | 228.17 (168) |
| TB cases detected | 3.07 (52) | 3.13 (52) | 4.25 (178) | 4.20 (178) |
| Average number of new COVID-19 cases over the study period per palika | 287 | | 1004 | |
| Average palika population size | 23 649 | | 46 007 | |

The average number of health visits per palika and month. The treatment group includes the palikas that lifted restrictions in 17 August 2020 to 16 September 2020 (Bhadra 2077). The control group includes those that maintained restrictions during that month. The (N) is the number of palikas reporting each month in the period for that service. National lockdown period (pre-period) includes 14 March 2020 to 15 July 2020 (Chaitra 2076 to Ashar 2077). The post-period includes 17 August 2020 to 16 September 2020 (Bhadra 2077). The average number of new COVID-19 cases in each period is COVID-19 cases at the district level since these data were not available at the palika level. The palika population size data were obtained from the Preliminary Data of National Population and Housing Census 2021.[39]
ANC, antenatal care; HIV, human immunodeficiency virus; PNC, postnatal care; TB, tuberculosis.

populations, on average, compared with the palikas in the control group. These differences are accounted for by the palika fixed effects in the DID design.

Figure 2 shows the trend in primary care service utilisation from January 2019 to September 2020 (equivalent to Magh 2075 to Bhadra 2077) and reveals parallel trends before and during the national lockdown period (our pre-period) for all services included. A sharp decrease in utilisation is observed in both groups of palikas at the start of the pandemic when the national lockdown was put in place (months 14–15 in figure 2). For most services, this decline was followed by a gradual resumption in the pre-period. Given these similar trends in both groups, the control palikas appear to provide appropriate counterfactual trends in the post-period. The joint F-test (online supplemental table 4) also did not reject the null hypothesis that the outcomes evolved differently in the treated vs the control palikas in the pre-period for the 10 services included. The parallel trend assumption was violated for two health services: visits for children under 5 with diarrhoea and pentavalent vaccinations, which were excluded from the analysis.

Estimates from DID regressions are reported in table 2 for the 10 health services. The coefficient for restrictions lifted is the DID estimate and can be interpreted as the difference in adjusted service utilisation between the treatment (lifted restrictions) and control (maintained restrictions) palikas in the post-period.

Lifting COVID-19 restrictions led to a positive increase in all services except total outpatient visits and PNC visits. These effects were statistically significant for three services. Lifting restrictions led to an average increase per palika of 57.5 contraceptive users (95% CI 14.6 to 100.5), 15.6 ANC visits (95% CI 5.3 to 25.9) and 1.6 child pneumonia visits (95% CI 0.2 to 2.9). Compared with the pre-COVID-19 average utilisation, this represented a 9.4% increase in contraceptive use, 34.2% increase in ANC visits and a 15.6% increase in child pneumonia visits.

Similarly, although not statistically significant, lifting restrictions led to 7.4 more children vaccinated against measles (95% CI −6.5 to 21.2), 5.0 more diabetes visits (95% CI −8.1 to 18.1), 12.7 more hypertension visits (95% CI −6.7 to 32.1), 34.8 additional HIV tests (95% CI −12.6 to 82.2) and 0.1 additional TB cases detected (95% CI −0.7 to 0.9) on average per palika. These increases were equivalent to increases of 8.7% for measles vaccinations, 10.7% for diabetes visits, 23.8% for hypertension visits, 39.8% for HIV tests and 2.0% for TB case detection in palikas that lifted restrictions compared with those that maintained them. In contrast, the coefficient for PNC and total outpatient visits were negative but these were not statistically significant. They were equivalent to

**Table 2** Estimated effect of lifting COVID restrictions on primary care service utilisation in Nepal, estimates from difference-in-differences models

| | Restrictions lifted | 95% CI | COVID-19 cases | 95% CI | N | R$^2$ | adj. R$^2$ |
|---|---|---|---|---|---|---|---|
| **Contraceptive users** | 57.51** | (14.55 to 100.48) | −0.01 | (−0.02 to 0.01) | 3710 | 0.01 | 0.01 |
| **ANC visits** | 15.60** | (5.34 to 25.86) | 0.01 | (−0.01 to 0.02) | 3625 | 0.08 | 0.07 |
| **PNC visits** | −1.50 | (−4.94 to 1.94) | 0.00 | (−0.00 to 0.00) | 2595 | 0.07 | 0.07 |
| **Child pneumonia visits** | 1.55* | (0.24 to 2.86) | −0.00** | (−0.00 to 0.00) | 2775 | 0.19 | 0.19 |
| **Measles vaccine** | 7.35 | (−6.49 to 21.19) | 0.00 | (−0.01 to 0.00) | 1740 | 0.20 | 0.20 |
| **Outpatient visits** | −56.81 | (−193.77 to 80.16) | −0.10*** | (−0.13 to 0.06) | 3710 | 0.09 | 0.09 |
| **Diabetes visits** | 5.01 | (−8.12 to 18.14) | 0.00 | (−0.01 to 0.02) | 1665 | 0.03 | 0.02 |
| **Hypertension visits** | 12.70 | (−6.74 to 32.14) | 0.00 | (−0.01 to 0.01) | 3445 | 0.02 | 0.02 |
| **HIV tests** | 34.83 | (−12.57 to 82.22) | 0.02 | (−0.02 to 0.05) | 1180 | 0.04 | 0.04 |
| **TB cases detected** | 0.06 | (−0.73 to 0.85) | 0.00 | (−0.00 to 0.00) | 1150 | 0.04 | 0.04 |

95% CI in parentheses.
The coefficient for restrictions lifted is the effect of lifting COVID restrictions on health service utilisation. Models also included fixed effects for month and palika.
*p<0.05, **p<0.01, ***p<0.001.
ANC, antenatal care; HIV, human immunodeficiency virus; PNC, postnatal care; TB, tuberculosis.

declines of 12.0% fewer PNC visits and 3.8% fewer outpatient visits in palikas that lifted restrictions compared with those that did not. Results from the sensitivity analysis that excluded 14 March 2020 to 12 April 2020 (Chaitra 2076) were largely consistent with the main model, with the exception of measles vaccinations which had a statistically significant increase in palikas that lifted restrictions (see online supplemental table 5).

## DISCUSSION

In this analysis, we used HMIS data and a DID design to estimate the effect of lifting COVID-19 restrictions on primary care health service utilisation in Nepal. We found that lifting restrictions increased contraceptive use, ANC and sick child visits by 9.4% to 34.2% on average across palikas. Utilisation of most other primary care services also increased by 2.0% to 39.8% but were not statistically significant. These results provide evidence that COVID-19 restrictions are linked to primary care service utilisation in Nepal and that lifting these restrictions can lead to an increase in service uptake. To our knowledge, this is the first paper to estimate the effect of lifting COVID-19 restrictions on health service utilisation using a quasi-experimental method.

There are many mechanisms through which COVID-19-related restrictions (stay-at-home requirements, business/workplace closures and public transport closures) might affect primary healthcare utilisation. People in Nepal have reported that public transport closures during the national lockdown prevented them from reaching healthcare facilities.[13] In addition, stay-at-home requirements meant that people were only permitted to leave their home for essential services. Although essential services included healthcare, stay-at-home requirements

were firmly enforced by law enforcement officials, and individuals were arrested and jailed or fined if they defied them.[14 24] Knowing this risk, people may have been deterred from seeking healthcare, despite being allowed to do so. It is also possible that many people did not know that visiting health facilities was allowed during the lockdown. The strict lockdown may have also increased anxieties around COVID-19 and deterred people from seeking care.[13 25] Once COVID-19 restrictions were lifted, these barriers would be subdued, and an increase in utilisation would be expected.

Lockdown policies, primarily stay-at-home requirements and business or workplace closures, can also have detrimental economic effects, potentially pushing low-income individuals and families further into poverty.[14 26] Most of the Nepali population works in the informal sector, including a large number in the tourism industry, which was severely impacted by the pandemic. Although the government developed economic support packages, informal sector workers or other marginalised groups often did not benefit from these.[14] Essential health services are supposed to be free of charge in public facilities in Nepal.[27] However, Nepalis often incur costs at point of service, with more than half of health expenditures being made out-of-pocket.[27 28] In addition, between 20% and 61% of people use private sector facilities for primary care in Nepal, depending on the type of service.[27 29 30] Lifting COVID-19 restrictions could have had an immediate effect on people's ability to generate income, especially for those working in the informal sector, and may have allowed them to pay for healthcare costs and thus seek health services again.

In this study, we found statistically significant effects only for reproductive, maternal and child health services

(RMNCH): contraceptive visits, ANC and child pneumonia visits. Both historically and during the pandemic, the government of Nepal has emphasised the promotion and improvement of RMNCH services. During COVID-19, RMNCH services were carefully monitored and mapped by Nepal's government to detect and address potential declines in coverage. The Country Preparedness and Response Plan focused heavily on maintaining RMNCH services and interim guidelines for RMNCH were also endorsed.[31] To our knowledge, no similar guidelines were issued for other primary care services. These important RMNCH-focused efforts might explain why effects were only statistically significant for these services and not for other, less promoted, services, such as noncommunicable diseases.[31] For example, policy makers suspended Measles-Rubella vaccinations campaigns, but aware of the risk of outbreaks, decided to continue the campaign after only 1 month. Nonetheless, we found no impact of lifting COVID-19 restrictions on facility-based PNC visits. This could be due to a large programme for PNC outreach (home visits) which was launched shortly before the pandemic and continued during the lockdown in some districts. This could explain why there was no difference in facility-based PNC between palikas that lifted restrictions and those that maintained them.

Other studies have shown that the declaration of the pandemic and the implementation of restrictions led to important declines in health service utilisation of varying magnitude and duration in many countries.[2–10] Studies from Nepal showed declines in primary care and hospital-based care following the implementation of COVID-19 restrictions including fewer deliveries and a potential increase in neonatal mortality and institutional stillbirths.[3 8] In contrast, our study assessed the effect of lifting these restrictions, and the resulting increase in service use using a DID design. DID designs compare trends between a treatment and comparison group and compare each group to itself, estimating an average of the counterfactual DID contrasts.

Nonetheless, our study has limitations. The DID design controls for time fixed differences between palikas (such as population size) and for secular trends affecting all groups. However, it is possible that remaining time-varying confounders affected the two groups differently. For example, although we adjusted for COVID-19 caseloads at the district level, it is possible that palika-specific outbreaks influenced the decision to maintain restrictions. Another limitation relates to the potential for measurement error for both the restrictions and the health service utilisation outcomes. The exposure variable may have been misclassified due to missing information on the restrictions in place in palikas. Although multiple sources were reviewed to collect and confirm the implementation of these restrictions, these sources sometimes lacked precision. Any misclassification due to missing information would have likely resulted in the palika mistakenly included in the treatment group (as having lifted the restrictions). This would bias the results towards the null. DHIS2 data are self-reported by facilities and may also contain errors, and reporting quality may have been affected by the pandemic. However, positive outliers were removed and only facilities that reported each indicator each month during the study period were included. It is unclear whether DHIS2 data quality issues would affect our analysis since misreporting should be similar in both the treatment and control groups. The estimates for TB case detection and measles vaccination must also be interpreted with caution as an important number of observations were excluded by the complete case analysis (see online supplemental table 6). In addition, our study was limited by the type of data available in the Nepal HMIS. For example, other important primary care services are not collected in the HMIS, such as mental health visits, which might have been affected by the pandemic. The Nepal HMIS also does not include data on home-visits by community health volunteers, which may be why we did not detect an increase in PNC visits, as described earlier. Furthermore, the Nepal HMIS only contains information aggregated at the health facility level, and we are unable to describe patient characteristics and demographic information. Finally, the outcome data were only available monthly, and the beginning and end of restrictions did not always match DHIS2 months precisely. Thus, policy dates and outcomes were not perfectly matched. In our dataset, the pre-period began 10 days before the national lockdown. However, sensitivity analyses that excluded the first month of the lockdown showed similar results (online supplemental table 5).

Our results have important implications for policy. We found that despite the ongoing COVID-19 pandemic, lifting restrictions can lead to an increase in RMNCH service utilisation. Universal utilisation of these services is crucial to improve health outcomes. ANC visits are essential to identify conditions that might threaten the mother or newborn's health.[32] It is estimated that a 10% decrease in coverage of pregnancy related and newborn healthcare during COVID-19 could result in an additional 28 000 maternal deaths and 168 000 neonatal deaths globally.[33] In addition, reduced contraceptive use could result in an increase in unintended pregnancies which can also place both the pregnant person and child at risk.[34] Delayed care for respiratory illnesses during COVID-19 restrictions could increase the incidence of pneumonia. Pneumonia is one of the leading causes of death for children under 5, and missed care could further exacerbate this burden.[35] Nonetheless, it is important to note that an increase in child pneumonia visits after restrictions were lifted could be linked to an increase in needs from further spread of respiratory illnesses (including COVID-19 and non-COVID-19) rather than from pent up demand.

Our study contributes to the literature on the indirect effects of COVID-19 restrictions on health systems. Although effective vaccines are now available, few people in LMICs are fully immunised against COVID-19 due to widespread inequities in access to vaccines.[36] Future waves of COVID-19 infections and emerging variants are

likely to push governments to consider reimplementing temporary restrictions and lockdowns. At the start of the pandemic, many countries took a one-size-fits-all approach with COVID-19 containment policies, as there was understandably much uncertainty surrounding COVID-19 and its effects. As we gain insight into the indirect effects of these restrictions, it is important that policy makers tailor these policies to their own demographic, disease and sociocultural contexts, and prepare health systems to respond accordingly.[26] Policy makers should consider strategies to promote and maintain all types of primary care services during future waves of COVID-19 and future pandemics. Such strategies may include better risk communication on the importance of essential healthcare and alternative service delivery modes such as telemedicine or differentiated service delivery strategies.[37] Health facilities should also be prepared to face potential increases in demand for healthcare when restrictions are eased. Strengthening public primary care services in Nepal is needed, including improving quality of care, and promoting better resilience during shocks.

**Contributors** NRK, CA, SB and MEK designed the study. NRK, CA, SM, AA and MD compiled and verified the data. NRK and CA led the data analysis. NRK wrote the first draft and all other coauthors, contributed to the interpretation of findings and read, improved and approved the final manuscript. NRK is the guarantor of the study.

**Funding** This work is part of the Grand Challenges ICODA pilot initiative, delivered by Health Data Research UK and funded by the Bill & Melinda Gates Foundation and the Minderoo Foundation (reference number 2021.0091). We also acknowledge funding from the Bill & Melinda Gates Foundation (grant INV-005254).

**Competing interests** None declared.

**Patient and public involvement** Patients and/or the public were involved in the design, or conduct, or reporting, or dissemination plans of this research. Refer to the Methods section for further details.

**Patient consent for publication** Not applicable.

**Ethics approval** This research was approved by the Nepal Health Research Council (NHRC), reference number 650, and determined to be exempt from a full review by the Institutional Review Board (IRB)of the Harvard T.H. Chan School of Public Health.

**Provenance and peer review** Not commissioned; externally peer reviewed.

**Data availability statement** Data may be obtained from a third party and are not publicly available.

**ORCID iD**
Neena R Kapoor http://orcid.org/0000-0001-8070-0995

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
