## [Reviewer comments · BMJ Open]

ARTICLE DETAILS

TITLE (PROVISIONAL)	The effect of lifting COVID-19 restrictions on utilization of primary care services in Nepal: a difference-in-differences analysis
AUTHORS	Kapoor, Neena; Aryal, Amit; Mehata, Suresh; Dulal, Mahesh; Kruk, Margaret; Bauhoff, Sebastian; Arsenault, C

VERSION 1 – REVIEW

REVIEWER	Shapira, Gil World Bank, Development Research Group
REVIEW RETURNED	19-Mar-2022

GENERAL COMMENTS	The authors estimate the effect of lifting COVID-19 restrictions on health service utilization in Nepal. Using difference-in-difference approach, they compare trends in service volumes in municipalities that lifted restrictions (treatment group) to those that kept enforcing restrictions (control group). The identification of the impacts relies on the assumption that if restrictions would not be lifted in the treatment group, then utilization would follow the same trend as in the control group. The authors find that lifting the restrictions caused an increase in three out of the 10 analyzed services, in a range of 9 to 34 percent. This study presents a contribution to the literature on the indirect impacts of the COVID-19 pandemic on health services. Several previous studies compare service volumes during periods with different levels of COVID-19 restrictions. The use of the DID approach, while not perfect, provides a more convincing estimation of the effect of restrictions on utilization. The authors are very transparent about the assumptions and the limitations of the approach. The analysis of data prior to the lifting of the restrictions supports the use of the approach. The paper is also very clear and well organized. I do, however, have a few comments that I hope can be helpful. 1. Which types of facilities report to the HMIS? Does it cover private facilities? Is data from hospitals included as well?2. Why are deliveries not included in the analysis? A study that found that there were fewer deliveries in Nepal during the pandemic is cited. Therefore, it seems important to test whether deliveries increased once restrictions were lifted. Delivery services are a good example of a service that cannot be delayed, unlike other services. For example, women can delay initiation of antenatal care rather than forgo care. Therefore, responsiveness to restrictions and their removal could be different.
--

	3. Can the three types of restrictions be separated? The current definition of restrictions combines stay-at-home orders, business/workplace closures, and limits on public transportation. I assume that workplace closures should have the least impact on service utilization and their inclusion might weaken the estimated impacts. If the data allows to separate the type of restrictions, the analysis could also provide clearer policy implications. 4. Restrictions are defined to be implemented in a municipality if any type of restriction was implemented for at least 10 days? Why was this threshold chosen? This implies that some treated municipalities could have no restrictions for most of the month, thus “diluting” the treatment. Have you tried a higher threshold? Alternatively, can you define the treatment as the number (or share) of days during a month in which restrictions were implemented? 5. Why did some municipalities choose to lift the restrictions while others didn’t? Is there any information on that in government documents or in the press? This could be helpful in terms of thinking of time-variant confounders. I also think it would be informative for readers to have a comparison of the two groups of municipalities in terms of their characteristics (population size, urban/rural,...) and number of COVID cases. 6. I do not understand the argument that reproductive, maternal and child health services are more resilient than other types of primary care services. Couldn’t the significant relationship between lifting the restriction and the volume of the services suggest that these services are more sensitive to the mobility restrictions? The paragraph in the Discussion section makes the case that these services are resilient because they rebounded. However, the paper doesn’t show which services were at all impacted by the pandemic and the initial introduction of the restrictions. 7. In the Conclusions section of the abstract, the authors write that the results call for policy makers to “...prepare health systems for potential rebounds in service utilization as restrictions are lifted.” I think it would be good to be clearer about what is meant by “rebound”. In what ways should health systems need to prepare? Does it imply that utilization increased above pre-pandemic levels when the restrictions are just removed?
--	--

REVIEWER	Das Sanyam, Sandip Sagarmatha Choudhary Eye Hospital, Cornea Research
REVIEW RETURNED	28-Jun-2022

GENERAL COMMENTS	Overall reflection: Although the authors have brought up a sensible context suggestive of common sense in health policy making, the conclusion is yet inconclusive. There is still scope to improve the quality of English, which is why it is recommended for further editing, preferably by a native speaker. It makes difficult for the reviewer to make specific comments in the specific section as line numbers are missing.
--

	The format of the date is not consistent throughout the manuscript. It is recommended to write the English date first followed by the Nepali date within parenthesis. Specific Comments: Abstract: Conclusion #1 Despite the ongoing pandemic, lifting restrictions can lead to an increase in reproductive, maternal, and child health service utilization. Our results call for policy makers in low- and middle-income countries to carefully consider the tradeoffs of strict lockdowns during future COVID waves or future pandemics and prepare health systems for potential rebounds in service utilization as restrictions are lifted. Comment: The service utilisation (average visit numbers) during the COVID lockdown seems higher as compared to post-lockdown. In what way is it said it may lead to an increase? Please justify how it will build trust with stakeholders and policymakers who believe this statement. Background: #2 On March 24, 2020, the Government of Nepal implemented a country-wide lockdown with strict restrictions on movement of people and closure..... Comment: This paragraph looks weird here. #3 Palika is referred as municipality government Comments: Palika is made up of both municipal and rural municipal governments. Methods: #4 These data were validated using the Nepal COVID Crisis Management Coordination Center (CCMCC) government websites.....additional online news sources..... reviewed by local researchers. Comment: news sources? Could you please be specific on this, which news portal/headlines/COVID update section and cite them with a date or date range? The authenticity of the news! (As per journal guidelines, these must be cited.)
--	---

	Who were the local researchers? Are they acknowledged? Potential conflict of interest!!! #5 We aimed to include 12 primary care services..... The following model was used and repeated for each of the 10 health service analyzed,..... Comment: The above two sentences are contradictory. What was the sampling method used in the study? Please clarify. If sampling was not done this must be highlighted within limitation. Were all the palikas included or there were some which were excluded? There doesn't seem any focus to this. Please clarify this in the methods. What was the reason for selecting 10 services? Were two services excluded? If so, please elaborate on why. Services like female community health volunteers (very active within the community) and mental health were not included. It would have been interesting to see that data as well, as there are multiple reports of mental health issues amid the COVID pandemic and lockdown. If these were not covered, then they must be included in the limitation segment. Results: #6 Table 1 shows t Comment: It is very unusual to see that the result starts directly with an illustration. Demographic information about patients who used health care services is lacking. Discussion: #7 Finally, pneumonia is one of the leading causes of death for children..... Comment: The common cold has a similar presentation to COVID initially. This may have led to postponing the visit during restriction and increasing the severity of the disease. Conclusion: Please rewrite the conclusion, as it may confuse readers from various fields.
--	---

	Figure:2 #8 Comment: It is surprising to see that despite restrictions, patient visits within palika are consistently higher. Please add panel numbers to all the figures separately. Most of the plots have a dip at the 15th month. There is no justification for this. Supplementary Table 1. #9 Comment: Abbreviations used within the table must not be familiar to readers from other backgrounds. suggested that it be expanded in the footnote Supplementary Table 2. #10 School closures – National¹ Comment: Superscript missing: National¹.
--	---

VERSION 1 – AUTHOR RESPONSE

Reviewers' Comments:

Reviewer #1:

The authors estimate the effect of lifting COVID-19 restrictions on health service utilization in Nepal. Using difference-in-difference approach, they compare trends in service volumes in municipalities that lifted restrictions (treatment group) to those that kept enforcing restrictions (control group). The identification of the impacts relies on the assumption that if restrictions would not be lifted in the treatment group, then utilization would follow the same trend as in the control group. The authors find that lifting the restrictions caused an increase in three out of the 10 analyzed services, in a range of 9 to 34 percent.

This study presents a contribution to the literature on the indirect impacts of the COVID-19 pandemic on health services. Several previous studies compare service volumes during periods with different

levels of COVID-19 restrictions. The use of the DID approach, while not perfect, provides a more convincing estimation of the effect of restrictions on utilization. The authors are very transparent about the assumptions and the limitations of the approach. The analysis of data prior to the lifting of the restrictions supports the use of the approach. The paper is also very clear and well organized.

I do, however, have a few comments that I hope can be helpful.

1. Which types of facilities report to the HMIS? Does it cover private facilities? Is data from hospitals included as well?

RESPONSE: *Thank you for this question. Both public and private facilities at all levels report to the Health Management Information System (HMIS) in Nepal, including hospitals. We added a sentence at lines 99 to 100 to ensure that this is clear and cited an Annual Report from the Department of Health Services, Ministry of Health and Population, Government of Nepal that describes this.*

(The line numbers provided here and those that follow are based on the unmarked manuscript.)

2. Why are deliveries not included in the analysis? A study that found that there were fewer deliveries in Nepal during the pandemic is cited. Therefore, it seems important to test whether deliveries increased once restrictions were lifted. Delivery services are a good example of a service that cannot be delayed, unlike other services. For example, women can delay initiation of antenatal care rather than forgo care. Therefore, responsiveness to restrictions and their removal could be different.

RESPONSE: *Thank you for this question. The focus of the paper was on primary care services, which is why we excluded deliveries (which are generally considered as part of tertiary care). To address the reviewer's question and because we had access to this data, we ran the analysis on the number of deliveries performed. We found no statistically significant effect from lifting restrictions. Looking at the graph below, deliveries appear to be increasing similarly in both groups of palikas (those that lifted restrictions and those that maintained them) in the post period.*

3. Can the three types of restrictions be separated? The current definition of restrictions combines stay-at-home orders, business/workplace closures, and limits on public transportation. I assume that workplace closures should have the least impact on service utilization and their inclusion might weaken the estimated impacts. If the data allows to separate the type of restrictions, the analysis could also provide clearer policy implications.

RESPONSE: *Thank you for this suggestion. Unfortunately, these policies were almost always implemented together which would prevent us from separating them in the analysis. In addition, the sources used to track the implementation and lifting of restrictions by palika were largely qualitative and occasionally lacked this level of precision.*

We revised the methods section to clarify the process used for tracking restrictions and creating the exposure variable (lines 151 to 161). We also added to the limitations section (lines 355 to 357) to address this point.

4. Restrictions are defined to be implemented in a municipality if any type of restriction was implemented for at least 10 days? Why was this threshold chosen? This implies that some treated municipalities could have no restrictions for most of the month, thus "diluting" the treatment. Have you tried a higher threshold? Alternatively, can you define the treatment as the number (or share) of days during a month in which restrictions were implemented?

RESPONSE: Thank you for these suggestions. Although this might be an interesting sensitivity analysis to add, as mentioned above, our sources for the types of policies in place and the dates of implementation and lifting occasionally lacked this level of precision. We have described the policy tracking process more clearly in the revised manuscript (lines 151 to 161) as well as clarified this in the limitations section (lines 355 to 360).

5. Why did some municipalities choose to lift the restrictions while others didn't? Is there any information on their government documents or in the press? This could be helpful in terms of thinking of time-variant confounders. I also think it would be informative for readers to have a comparison of the two groups of municipalities in terms of their characteristics (population size, urban/rural,...) and number of COVID cases.

RESPONSE: Thank you for this question. According to news sources and local co-authors, districts or palikas that implemented restrictions after the national lockdown was lifted often did so because there were COVID cases found in the palika. This is why we chose to control for the number of new COVID cases.

To address the reviewer's comment, we revised **table 1** and added the average palika population size and number of COVID cases in the treated vs. control groups (line 221).

In addition to the average number of visits per palika and month for each service, these data indicate that palikas in the treated group tend to have smaller populations, less COVID and serve fewer patients. They are therefore more likely to be rural.

We commented on this in the results and discussion sections (lines 216 to 218 and 348 to 352).

6. I do not understand the argument that reproductive, maternal and child health services are more resilient than other types of primary care services. Couldn't the significant relationship between lifting the restriction and the volume of the services suggest that these services are more sensitive to the mobility restrictions? The paragraph in the Discussion section makes the case that these services are resilient because they rebounded. However, the paper doesn't show which services were at all impacted by the pandemic and the initial introduction of the restrictions.

RESPONSE: Thank you for bringing up these points. We agree that this argument might not be appropriate given our study design and have revised this. We instead highlighted the fact that many interventions implemented by the Government of Nepal and partners during COVID have focused on maintaining maternal and child health services specifically (lines 325 to 333). This might explain the increase in utilization of these services after restrictions were lifted, while other services were not directly targeted by these interventions.

We also cited a previous study in the Background and discussion sections (lines 321 to 323) that estimated the magnitude of disruptions in several services following the declaration of the pandemic and found that MCH services were less affected than other services like NCDs.

7. In the Conclusions section of the abstract, the authors write that the results call for policy makers to "...prepare health systems for potential rebounds in service utilization as restrictions are lifted." I think it would be good to be clearer about what is meant by "rebound". In what ways should health systems need to prepare? Does it imply that utilization increased above pre-pandemic levels when the restrictions are just removed?

RESPONSE: Thank you for highlighting this. We agree that "rebound" might not be the appropriate term and have changed this to "potential increases" (line 27). We do know that these services were disrupted at the start of the pandemic, as mentioned above, but they do not appear to have surpassed pre-pandemic levels after restrictions were lifted.

Reviewer #2:

Overall reflection:

Although the authors have brought up a sensible context suggestive of common sense in health policy making, the conclusion is yet inconclusive.

RESPONSE: Thank you for highlighting this important point. We have revised the conclusion section. Please see answers to comments 6 and 7 of reviewer 1.

There is still scope to improve the quality of English, which is why it is recommended for further editing, preferably by a native speaker.

RESPONSE: Thank you. Co-authors have reviewed the text and edited any grammatical errors.

It makes difficult for the reviewer to make specific comments in the specific section as line numbers are missing.

RESPONSE: Thank you. We have now added line numbers in hopes that this makes the review process easier. The line numbers that follow are based on the unmarked manuscript.

The format of the date is not consistent throughout the manuscript. It is recommended to write the English date first followed by the Nepali date within parenthesis.

RESPONSE: Thank you for pointing this out. We have updated this throughout the manuscript where appropriate.

Specific Comments:

Abstract: Conclusion

#1 Despite the ongoing pandemic, lifting restrictions can lead to an increase in reproductive, maternal, and child health service utilization. Our results call for policy makers in low- and middle-income countries to carefully consider the tradeoffs of strict lockdowns during future COVID waves or future pandemics and prepare health systems for potential rebounds in service utilization as restrictions are lifted.

Comment: The service utilisation (average visit numbers) during the COVID lockdown seems higher as compared to post-lockdown. In what way is it said it may lead to an increase? Please justify how it will build trust with stakeholders and policymakers who believe this statement.

RESPONSE:

*Thank you for this question. According to **table 1**, the average number of visits per palika and month is generally lower in the pre-period compared to the post-period, although there is some variability. For example, in the pre-period, palikas in the treatment group conducted an averaged of 88 HIV tests compared to 126 in the post period.*

*Nonetheless, the main findings of an increase in some services after lifting the restrictions are obtained using a difference-in-differences model where we adjust for time-fixed confounders and secular trends. We're confident that these methods are appropriate and the main findings in **table 2** allow us to drathese conclusions.*

To improve interpretability for stakeholders and policy makers we revised the conclusion section and have also added a "Strengths and Limitations of the study" section at the beginning of the manuscript as requested by the editor.

Background:

#2 On March 24, 2020, the Government of Nepal implemented a country-wide lockdown with strict restrictions on movement of people and closure.....

Comment: This paragraph looks weird here.

RESPONSE: *We have rephrased these sentences to make the paragraph clearer (lines 81 to 88).*

#3 Palika is referred as municipality government

Comments: Palika is made up of both municipal and rural municipal governments.

RESPONSE: *Thank you for mentioning this. We have now updated this in the background section to: "753 urban and rural municipalities" (line 85).*

Methods:

#4 These data were validated using the Nepal COVID Crisis Management Coordination Center (CCMCC) government websites.....additional online news sources..... reviewed by local researchers.

Comment:

1. news sources? Could you please be specific on this, which news portal/headlines/COVID update section and cite them with a date or date range? The authenticity of the news! (As per journal guidelines, these must be cited.)

RESPONSE: *Thank you for noting this. We revised the methods section to clarify the process used to compile the data on restrictions implemented and lifted per palika (lines 151 to 161). We have cited*

the primary sources in the manuscript and additional news sources used in **supplementary material table 1** for improved transparency and reproducibility.

2. Who were the local researchers? Are they acknowledged? Potential conflict of interest!!!

RESPONSE: Thank you for noting this. The local researchers are co-authors of this paper. Three of the seven co-authors are Nepali researchers or policy makers. In the revised methods (line 153) we clarified that the local researchers are co-authors. The first author did the first extraction which was revised thoroughly by the three Nepali co-authors.

#5 We aimed to include 12 primary care services.....

The following model was used and repeated for each of the 10 health service analyzed,.....

Comment: The above two sentences are contradictory.

RESPONSE: Thank you for catching this inconsistency. Two services had to be excluded due to the violation of the parallel trends assumption required for difference in differences estimation. We have clarified this in the revised text (line 173 and lines 241 to 242).

1. What was the sampling method used in the study? Please clarify. If sampling was not done this must be highlighted within limitation.

RESPONSE: We used data from the DHIS2 platform which is expected to include data from all health facilities in Nepal. We have reorganized the methods so that this is clearer and mention the number of facilities reporting to the DHIS2 at lines 100-101. We also mention the limitations of these data in the discussion section.

2. Were all the palikas included or there were some which were excluded? There doesn't seem any focus to this. Please clarify this in the methods.

RESPONSE: Thank you for this comment. Of the 753 palikas in Nepal, only one was fully excluded from the analysis because it did not report consistently over the 5-month period for any of the 10 services analyzed. However, not all palika report on all services and some palika report inconsistently. For each health service analyzed, we included only palikas that reported data for all 5 months of the study period (complete case analysis). Thus, the number of palikas included in the dataset varies by health service and are presented in **table 1**. In **supplemental table 6**, we included the number of palikas and sum of services included in the dataset before and after the complete case analysis. Less than 8% of services were excluded for all services except TB case detection and measles vaccination. We added this in the limitations section (lines 363 to 365).

3. What was the reason for selecting 10 services? Were two services excluded? If so, please elaborate on why.

RESPONSE: Thank you for asking this question. We aimed to include all primary care services we could extract from the DHIS2 platform. At lines 241-242 of the manuscript, we mention that two services were excluded because they violated the parallel trends assumption. The trends for these two services in the pre-period were not parallel based on our statistical test, thus, differences-in-differences analysis would not be appropriate for these services.

4. Services like female community health volunteers (very active within the community) and mental health were not included. It would have been interesting to see that data as

well, as there are multiple reports of mental health issues amid the COVID pandemic and lockdown. If these were not covered, then they must be included in the limitation segment.

RESPONSE: Thank you for mentioning these important services. Unfortunately, we were limited by the services available in the Nepal DHIS2. We have now added this as a limitation (lines 366 to 368).

Results:

#6 Table 1 shows t

Comment:

1. It is very unusual to see that the result starts directly with an illustration.

RESPONSE: We revised this section in the manuscript to present the other text first (line 213).

2. Demographic information about patients who used health care services is lacking.

RESPONSE: Thank you for noting this. The health service utilization data currently available in the Nepali DHIS2 are only at the palika or health facility level and we do not have any information on the patients using these services. Nonetheless, in the revised **table 1**, we added information on the demographic characteristics of the palikas, notably the population sizes and number of COVID cases.

Discussion:

#7 Finally, pneumonia is one of the leading causes of death for children.....

Comment: The common cold has a similar presentation to COVID initially. This may have led to postponing the visit during restriction and increasing the severity of the disease.

RESPONSE: Thank you for noting this. We have added a sentence that highlights this point (381-382).

Conclusion: Please rewrite the conclusion, as it may confuse readers from various fields.

RESPONSE: Thank you. As described in the first comment, we have revised the conclusion in the abstract and the discussion section. In addition, we believe that the reorganized and revised methods, the added characteristics of palikas in **table 1**, and the additional information on the number and types of health facilities analyzed, will help improve clarity.

Figure:2

#8

Comment:

1. It is surprising to see that despite restrictions, patient visits within palika are consistently higher.

RESPONSE: Thank you for noting this. This is likely because the palikas that maintained the restrictions in place were generally larger palikas. As described in the revised **table 1**, the control palikas that maintained restrictions had higher average health service visits, larger populations and more COVID cases.

We describe at lines 218-219 that the palika fixed effects in the DID design will account for these differences across the two groups.

2. Please add panel numbers to all the figures separately.

RESPONSE: Thank you. We added letters to each panel in figure 2.

3. Most of the plots have a dip at the 15th month. There is no justification for this

RESPONSE: Thank you for highlighting this point. The drop at month 15 (March 2020) is when the COVID-19 pandemic was declared globally, and the national restrictions were initially put in place. We have added a sentence in the results section to highlight this point (lines 234 to 237).

Supplementary Table 1.

#9

Comment: Abbreviations used within the table must not be familiar to readers from other backgrounds. suggested that it be expanded in the footnote

RESPONSE: Thank you for bringing this to our attention. These variable names are extracted directly from DHIS2 as written, so we have added the abbreviations to the footnote as you suggested.

Supplementary Table 2.

#10

School closures – National¹

Comment: Superscript missing: National¹.

RESPONSE: Thank you for identifying this. We have now updated it in the **supplementary materials**.

VERSION 2 – REVIEW

REVIEWER	Shapira, Gil World Bank, Development Research Group
REVIEW RETURNED	11-Aug-2022
GENERAL COMMENTS	I think that the revisions improved the manuscript and that the authors appropriately addressed all comment.
REVIEWER	Das Sanyam, Sandip Sagarmatha Choudhary Eye Hospital, Cornea Research
REVIEW RETURNED	28-Aug-2022

GENERAL COMMENTS	Abstract and main Conclusion: The conclusion still says that: "Our results call for policymakers in low- and middle-income countries to carefully consider the tradeoffs of strict lockdowns during future COVID waves or future pandemics and prepare health systems for potential increases in service utilisation as restrictions are lifted." But, given that the results show that service utilisation increased primarily during the COVID restrictions, how can the authors conclude that "health systems should prepare for potential increases in service utilisation as restrictions are lifted?" In common sense, palika health centres are in the community, and if there are restrictions on travel, then people are bound to visit there, but if the restrictions are eased, they go to seek service at higher centres. This is the reason why there was an increase when there was a restriction. Suggestion: Conclude by strengthening community health systems so that they are able to provide standard care despite hard times like the pandemic. Introduction: Why is all this information in line numbers 81-88 in the introduction? description of palika and things that can go into methodology. Methods: Line 116: Comment: It is still unclear why this number of services was included. Aim says 12 but the data says 10. Elaborate on the question below as well. What was the reason for selecting 10 services? The below quoted comment is not addressed. Services like female community health volunteers (very active within the community) and mental health were not included. It would have been interesting to see that data as well, as there are multiple reports of mental health issues amid the COVID pandemic and lockdown. If these were not covered, then they must be included in the limitation segment. Result: Comment: Demographic information about patients who used health care services is still missing. If this was not collected, why is it not mentioned in the limitation? Epidemiologists would benefit greatly from demographic information on health-care seeking behavior.
--

VERSION 2 – AUTHOR RESPONSE

Reviewers' Comments:

Reviewer: 1

Dr. Gil Shapira, World Bank

Comments to the Author:

I think that the revisions improved the manuscript and that the authors appropriately addressed all comment.

RESPONSE: *Thank you for your comments and feedback.*

Reviewer: 2

Mr. Sandip Das Sanyam, Sagarmatha Choudhary Eye Hospital

Comments to the Author:

Abstract and main Conclusion:

The conclusion still says that:

"Our results call for policymakers in low- and middle-income countries to carefully consider the tradeoffs of strict lockdowns during future COVID waves or future pandemics and prepare health systems for potential increases in service utilisation as restrictions are lifted."

But, given that the results show that service utilisation increased primarily during the COVID restrictions, how can the authors conclude that "health systems should prepare for potential increases in service utilisation as restrictions are lifted?"

In common sense, palika health centres are in the community, and if there are restrictions on travel, then people are bound to visit there, but if the restrictions are eased, they go to seek service at higher centres. This is the reason why there was an increase when there was a restriction.

Suggestion: Conclude by strengthening community health systems so that they are able to provide standard care despite hard times like the pandemic.

RESPONSE: *Thank you for this question and comment. Our paper does not aim to describe service trends during the Covid-19 pandemic. Even though services increased during and after the national lockdown (because most of them quickly re-bounded after the initial pandemic-related drop), what our paper shows is the difference in effects between the 'treated' and 'control' palikas when the national lockdown was lifted in Nepal. We found that palikas that lifted restrictions had a statistically significant greater increase in utilization for several services compared to palikas that maintained restrictions. Although both groups may be "increasing" in the post period, the data shows that treated palikas (those that lifted restrictions), increased more than control palikas (those that maintained restrictions). Our result therefore points to a causal effect*

of lifting restrictions on service use. We modified the last sentence of the conclusion section in the abstract on lines 25-26 (Note: line numbers are based on the version without track changes).

We invite the reviewer to consult the following texts which describes the use of difference-in-differences methods:

Angrist JD, Pischke J-S. 2008. Parallel worlds: fixed effects, difference-in-differences, and panel data. Mostly Harmless Econometrics. An Empiricist's Companion. Princeton, NJ: Princeton University Press, 221–47.

<https://www.publichealth.columbia.edu/research/population-health-methods/difference-difference-estimation>

We have added one sentence explaining the DID design further on lines 172-174.

In addition, our data included all health facilities in Nepal including community health centers as well as higher-level centers. A switch in facility types would therefore not affect our study as the total number of services provided in the country as a whole would remain the same.

We appreciate the reviewer's comment that community health centers should be strengthened to provide essential care during pandemics or other shocks. We have added a sentence on this in the main text conclusion at lines 410-412.

Introduction:

Why is all this information in line numbers 81-88 in the introduction? description of palika and things that can go into methodology.

RESPONSE: *Thank you for this comment. Based on the reviewer's suggestion, we have shortened this paragraph in the introduction and moved the information to the methods section.*

Methods:

Line 116: Comment: It is still unclear why this number of services was included. Aim says 12 but the data says 10. Elaborate on the question below as well.

What was the reason for selecting 10 services?

RESPONSE: Thank you for this comment. We aimed to include all 12 primary care services. In the results, we explain that two services (visits for child diarrhea and pentavalent vaccinations) had to be excluded due to violation of the parallel trends assumption, which is a requirement for difference-in-differences estimation (lines 243 to 245). If trends in an outcome are not parallel in the pre-intervention period, the control group is not a good counterfactual for the treatment group in the post period. Therefore, these two services cannot be analyzed using a DID design.

In the supplementary materials, we provide a table with the results of the F-test (a statistical test to test the parallel trend assumption, supplementary table 4).

In the methods (lines 118-121), we also added a sentence describing the selection of these 12 services.

The below quoted comment is not addressed.

Services like female community health volunteers (very active within the community) and mental health were not included. It would have been interesting to see that data as well, as there are multiple reports of mental health issues amid the COVID pandemic and lockdown. If these were not covered, then they must be included in the limitation segment.

RESPONSE: We apologize for missing this comment in the first revision. We agree these services would be very interesting to study, but unfortunately, they are not included in the Nepali DHIS2. In the Discussion section we had previously mentioned the limitations of the Nepali DHIS2 and that it does not include data on several important services including mental health care (lines 369-370).

Based on the reviewer's suggestion, we added a sentence on the fact that home-visits from community volunteers were not included (lines 370-372).

Comment: Demographic information about patients who used health care services is still missing. If this was not collected, why is it not mentioned in the limitation?
Epidemiologists would benefit greatly from demographic information on health-care seeking behavior.

RESPONSE: Thank you for noting this. The Nepali DHIS2 only contains information aggregated at the health facility level (e.g., total volume of services provided any given month). It does not contain any patient-level information.

We have added a sentence on this as a limitation at lines 372-374.

In the first revision, we aimed to partially address this comment by adding the average population size in the treated vs. control palikas as well as the average number of new Covid cases (**Table 1**). This shows that our treated groups tend to include smaller municipalities and faced fewer COVID cases.